# Structural Abnormalities in Brugada Syndrome and Non-Invasive Cardiac Imaging: A Systematic Review

**DOI:** 10.3390/biology12040606

**Published:** 2023-04-17

**Authors:** Martina De Raffele, Assunta Di Domenico, Cristina Balla, Francesco Vitali, Alberto Boccadoro, Rita Pavasini, Marco Micillo, Marta Cocco, Gianluca Campo, Matteo Bertini, Elisabetta Tonet

**Affiliations:** Cardiovascular Institute, Azienda Ospedaliero-Universitaria di Ferrara, 44124 Cona, FE, Italy

**Keywords:** Brugada syndrome, echocardiography, cardiac magnetic resonance, cardiomyopathy

## Abstract

**Simple Summary:**

Brugada syndrome (BrS) has always been considered a purely electrical disease and imaging techniques do not currently play a specific role in the diagnosis of this arrhythmic syndrome. The aim of this review is to identify possible structural abnormalities of BrS and their potential association with symptoms, risk stratification, and prognosis.

**Abstract:**

The aim of this review is to identify possible structural abnormalities of BrS and their potential association with symptoms, risk stratification, and prognosis. (1) Background: BrS has always been considered a purely electrical disease and imaging techniques do not currently play a specific role in the diagnosis of this arrhythmic syndrome. Some authors have recently hypothesized the presence of structural and functional abnormalities. Therefore, several studies investigated the presence of pathological features in echocardiography and cardiac magnetic resonance imaging (MRI) in patients with BrS, but results were controversial. (2) Methods: We performed a systematic review of the literature on the spectrum of features detected by echocardiography and cardiac MRI. Articles were searched in Pubmed, Cochrane Library, and Biomed Central. Only papers published in English and in peer-reviewed journals up to November 2021 were selected. After an initial evaluation, 596 records were screened; the literature search identified 19 relevant articles. (3) Results: The imaging findings associated with BrS were as follows: right ventricular dilation, right ventricular wall motion abnormalities, delayed right ventricular contraction, speckle and feature tracking abnormalities, late gadolinium enhancement, and fat infiltration in the right ventricle. Furthermore, these features emerged more frequently in patients carrying the genetic mutation on the sodium voltage-gated channel α-subunit 5 (SCN5A) gene. (4) Conclusions: Specific imaging features detected by echocardiography and cardiac magnetic resonance are associated with BrS. However, this population appears to be heterogeneous and imaging anomalies emerged to be more frequent in patients carrying genetic mutations of SCN5A. Future studies with an evaluation of BrS patients are needed to identify the specific association linking the Brugada pattern, imaging abnormalities and their possible correlation with prognosis.

## 1. Introduction

Brugada syndrome (BrS) is an inherited cardiac disorder associated with an increased risk of ventricular tachyarrhythmias and sudden cardiac death in young and healthy adults. The worldwide prevalence of BrS is estimated to be 0.05%, it is more common in young males with a prevalent geographic/ethnic distribution in Asian countries [1,2].

Inheritance of BrS is via an autosomal dominant transmission with incomplete penetrance. Despite more than 20 different genes having been proposed as being implicated in BrS, mutations in the SCN5A (sodium voltage-gated channel α-subunit 5) gene located on chromosome 3 account for 20–30% of patients with Brugada syndrome [2,3].

The diagnosis of BrS is based on a typical electrocardiographic pattern characterized by ST-segment elevation >2 mm followed by a concave or straight ST segment with a negative symmetric T-wave in at least one right precordial lead (V1 and/or V2), placed in a standard or superior position (up to the 2nd intercostal space). This pattern can occur either spontaneously or after provocative drug test with intravenous administration of sodium-channel blockers [1,4,5].

BrS can demonstrate a wide spectrum of clinical manifestations, ranging from asymptomatic patients to patients experiencing severe symptoms such as syncope or aborted sudden cardiac death due to polymorphic ventricular tachycardia and ventricular fibrillation [6]. The first studies reported a rate of syncope or sudden cardiac death of about 17% and 42%, respectively. This number might overestimate the true incidence of these events, due to the high percentage of asymptomatic patients who remain undiagnosed. Recent studies reported a significantly lower percentage of sudden cardiac death as the first symptom and a lower incidence of recurrent arrhythmia during follow-up, which are both around 5% [7].

Initially considered as a purely electrical disease, using cardiac imaging just to exclude other cardiomyopathies, some authors have recently hypothesized the presence of structural and functional abnormalities, mostly located in the right ventricle (RV) and, specifically, in the outflow tract [8]. Therefore, several studies have investigated the presence of pathological features in echocardiogram and magnetic resonance in patients bearing BrS, but with conflicting results. The present systematic review is carried out to identify and to describe the spectrum of features detected by transthoracic echocardiography (TTE) and cardiac magnetic resonance (CMR) in BrS patients.

## 2. Methodological Considerations

To summarize current evidence regarding the imaging techniques and BrS, a systematic review in accordance to Preferred Reporting Items for Systematic reviews and Meta-Analyses (PRISMA) guidelines was performed [9]. The terms searched were: “((cardiac imaging) OR (echocardiography) OR (cardiac magnetic resonance) OR (CMR) OR (cardiac MRI)) AND ((BrS) OR (Brugada Syndrome) OR (Brugada pattern))”. The databases analyzed were PubMed, Biomed Central, and Cochrane Library. Only papers published in English and in peer-reviewed journals were selected. The inclusion criteria of the studies were: observational or clinical trials involving patients with BrS evaluated with cardiac magnetic resonance (CMR) or echocardiography. Overall, 845 studies were selected (Figure 1). After a first evaluation, 845 records were screened, and of these 667 were excluded for different reasons (Figure 1). Finally, 19 studies were included in the systematic review. The quality of the included study has been tested using pre-specified electronic forms of MINORS criteria [10]. The minimum score obtained was 10 and the maximum was 20. No studies were excluded on the basis of quality assessment. We presented results of the systematic review, answering questions related to the imaging findings in BrS.

## 3. Cardiac Imaging in BrS Patients

There are no strict indications about cardiac imaging in BrS patients. Taking into account symptoms and the possible association with RV morphological abnormalities, echocardiography represents the first choice for structural pathological features detection. Therefore, TTE can be considered as mandatory in BrS subjects. However, current guidelines recommend CMR as the gold standard for the assessment of biventricular dimension and function [11]. Additionally, CMR allows tissue characterization in order to identify pathological myocardial composition.

The present systematic analysis of literature included 9 studies with TTE analysis of BrS patients and 10 studies with CMR investigation.

The following structural abnormalities were detected: RV dilatation, abnormal RV ejection fraction (EF), altered speckle and feature tracking, RV contraction delay, abnormal tissue characterization, and atrial involvement.

If the technique for detection of each abnormality was TTE or CMR was showed in Table 1, the estimated incidence of each parameter was reported in Table 2.

Each one of these features is separately described below.

### 3.1. Right Ventricular Dilatation

Several studies compared the echocardiographic characteristics of patients with BrS and healthy individuals. On one hand, there are studies that support the absence of structural alterations in patients with BrS [12]. On the other hand, it was found that BrS patients showed an increase in right ventricular size, especially in the outflow tract diameter (RVOT) and telediastolic (RVED) and telesystolic (RVES) volumes with statistically significant impact [12,13,14]. In order to exclude overlap syndromes between BrS and cardiomyopathies such as arrhythmogenic right ventricular cardiomyopathy (ARVC), some studies analyzed BrS and ARVC patients. In particular, a study by Iacoviello M et al. was conducted comparing patients with BrS, patients with ARVC and healthy controls. The comparison of two-dimensional and Doppler measurements of left ventricle (LV) and RV between BrS patients and control group did not show any differences. In contrast, patients with ARVC showed significantly higher LVED volumes, as well as distal and proximal RVOT, basal RV diameter, end-diastolic and end-systolic RV areas compared to BrS patients and controls [15].

Taking into account the intra and inter-operator variability of echocardiography, several studies analyzed RV dimension by CMR.

CMR studies regarding BrS produced conflicting results in terms of RV enlargement. Some studies showed increased ventricular dimensions [16,17,18,19,20,21,22] associated with reduced ventricular function not confirmed in other data from literature [21]. Table 3 summarizes all findings of studies about RV dimensions in BrS patients. On the basis of the concept that BrS could be a disease of RVOT, some authors focused on the dimension of this RV tract. Gray B et al. showed that BrS patients exhibited normal global RV volumes but larger RVOT volumes than controls [21]. Papavassilu T et al. compared CMR images of 69 BrS patients with that of 30 controls demonstrating a significantly enlarged RVOT in patients with spontaneous type 1 ECG [8] (Figure 2).

### 3.2. Ejection Fraction

Ejection fraction (EF) represents one of the most important imaging parameters due to its implication in terms of symptoms, arrhythmias, and management strategies. CMR is considered the gold standard for the assessment of EF [11]. The majority of studies assessing EF in BrS patients showed an impaired EF [8,16,18,19].

The first prospective analysis encompassing CMR in BrS patients was performed by Papavassiliu et al., who enrolled 20 BrS compared to matched controls [19]. A trend to a lower RVEF in patients with BrS was noticed, even though the difference did not reach statistical significance. No significant difference was found in left ventricular (LV) parameters between the two cohorts.

Analyzing a cohort of 30 BrS patients, Catalano et al. described the presence of RV contractility abnormalities in a relevant percentage of BrS patients (50%). In particular, a significantly reduced contractility in the anterior-apical segment and in the RVOT was detected and a borderline association with reduced contractility in the inferior mid-ventricular segment in the inflow tract was found [16].

Another study published by Papavassiliu et al. compared CMR of 30 BrS patients (both with spontaneous and drug induced pattern) with healthy controls. Irrespective of the presence of spontaneous or induced ECG pattern, BrS patients showed lower RVEF compared to healthy controls. Interestingly, patients with a spontaneous ECG pattern also exhibited LV functional impairment, specifically a lower LVEF and thicker posterior wall. Spontaneous BrS pattern patients revealed biventricular functional and morphological alterations in comparison to patients with drug induced BrS pattern patients and healthy controls, respectively [8].

Taking into account these data, three studies afterward investigated the possible correlation between CMR findings and genetic mutations, with the aim to compare genotypical and phenotypical features. In particular, they demonstrated an impaired wall motion and EF of the RV in patients with SCN5A mutation [18,21,22]; interestingly, in subjects carrying SCN5A mutation, the study by Rudic et al. also showed a trend towards increased LV diameter and lower LVEF when compared with healthy controls. Among these studies, Gray et al. suggested that patients with BrS had preserved overall RV function, but evident abnormalities were confined to the RVOT; these abnormalities were related to age and rare genetic variants. Moreover, a significant correlation between spontaneous type 1 ECG changes and subtle RV dysfunction (lower RVEF) emerged; these data were also demonstrated by Rudic et al. [18].

In a recent study by Hohneck et al., CMR and CMR feature tracking analysis were performed on a cohort of BrS patients, both spontaneous BrS and drug induced BrS patterns, compared with matched healthy controls. Authors underlined a trend to lower biventricular EF in patients with spontaneous BrS pattern when compared with subjects with drug induced BrS pattern and healthy controls, respectively, even if the difference did not reach statistical significance [20].

Conflicting results were reported by two other studies: on one hand, Von Hoorn et al. showed that patients carrying SCN5A mutation had significantly lower LVEF than non-SCN5A-BrS patients and healthy controls, without a difference in RVEF [17]. On the other hand, Tessa et al. demonstrated that RV wall motion abnormalities detected in BrS patients were also found in healthy subjects, possibly representing physiological features. Therefore, the authors concluded that BrS might occur in structurally and functionally normal heart [23]. It has to be noted that genetic typing and history of major arrhythmic events were lacking in the population analyzed by Tessa et al.; therefore, the absence of imaging abnormalities could be explained. Table 4 summarizes findings from studies about EF in BrS patients.

### 3.3. Speckle and Feature Tracking

In the last few years, RV strain imaging emerged as a fundamental tool to estimate RV systolic performance, overcoming some limitations of conventional echocardiographic parameters [24]; indeed, speckle-tracking echocardiography (STE) techniques are independent from the angle of insonation, and offer greater reproducibility than conventional echocardiography [25]. Strain parameters can be also performed analyzing CMR images by feature tracking technique. This technique may be useful in the early detection of myocardial functional impairments of several pathologies, including the ones involving RV wall motion abnormalities [26,27,28]. Moreover, this tool might be useful for risk stratification, both in patients with ischemic and non-ischemic cardiomyopathies [29,30,31,32].

With this background, several studies assessed strain parameters in BrS subjects. Abnormal strain imaging (echo and CMR) was found in BrS patients with significantly lower values when compared to healthy controls, even if these findings were shown to be less evident than the ones observed in arrhythmogenic cardiomyopathy [20,33] (Figure 3).

Mitroi C et al. investigated the relationship between some strain RV parameters and arrhythmic events in BrS patients: RV STE abnormalities (RIMP > 0.50, RVOTS < 16.2% and RVMDm > 42 ms) resulted to be sensitive predictors of arrhythmic events after a 7-years follow-up. Hence, the authors underlined the importance of assessing RVOT contraction time to identify high-risk patients, because RVOT mechanical substrate might demonstrate a strict correlation with arrhythmic risk [33].

A prospective study by Hohneck et al. evaluated the prognostic impact of CMR feature tracking in a well characterized cohort of BrS patients, to detect subclinical impairment and identify those with higher risk for arrhythmogenic events. CMR feature tracking showed remarkable differences in RV strain parameters between healthy controls and patients with spontaneous BrS pattern, while conventional CMR measures were comparable. On the contrary, no CMR feature tracking difference was found among patients with drug induced BrS pattern and healthy population. Regarding follow-ups, major adverse event occurred mainly in patients with spontaneous BrS pattern. The strongest correlation with arrhythmic events was found with RV global circumferential strain (GCS) and RVOT size. Therefore, the authors emphasized the clinical importance of CMR feature tracking as a reproducible tool, which might be useful to assess high risk patients [20].

### 3.4. Contraction Delay

Ventricular ejection delay is defined as the time from QRS onset to the onset of the systolic ejection wave at the end of the isovolumetric contraction. It represents the conduction delay and the electromechanical coupling for each ventricular wall. Some studies tried to evaluate these parameters in BrS subjects. Mitroi C et al. found that BrS patients exhibited a greater contraction delay between the lateral and the septal aspect of the RV when compared to controls [33]. Van Malderen SCH et al. analyzed contraction delay in BrS individuals and controls and its relationship with arrhythmias: the study results showed that BrS patients had longer RV ejection delay and interventricular ejection delay than controls and that these parameters were the longest ones in BrS patients with a previous history of syncope or spontaneous ventricular arrhythmia. An analysis of differences among the three Brugada patterns showed that type 1 Brugada patients had a longer ejection time when compared with type 2 and 3 Brugada patients and controls. Among type 1 Brugada patients, ejection time was significantly greater in patients who had previous cardiac events: in the presence of type 1 Brugada ECG pattern, an ejection time of 40 ms identified patients likely to have cardiac events, with 85.7% sensitivity and 93.7% specificity [34]. Considering the hypothesis of RVOT abnormalities in BrS, Trevisan et al. used DTI to assess whether time to peak myocardial shortening at the RVOT is prolonged in comparison to that at the RV inflow tract. Taking into account the dynamic nature of type 1 Brugada pattern, they analyzed patients during type 1 pattern expression. They showed that during spontaneous type 1 Brugada pattern expression, there was a contraction delay at the RVOT level which was not evident in patients without expression of Brugada pattern at the time of TTE [13].

### 3.5. Tissue Carachterization

Several MRI studies investigated the presence of myocardial tissue abnormalities in BrS patients: in particular, they focused on myocardial fibrosis and fat infiltration [15,19,20,21,23]. Regarding LGE, the majority of CMR studies did not find fibrosis affecting LV or RV of BrS patients [21,22]. An analysis by Bastiaenen et al. showed LGE in the midwall of the left ventricle in 8% of 78 BRS patients and in RV insertion point, but not in the RV myocardium [22]. Previous literature data focused on fatty infiltration highlighting conflicting results. On one hand, a study by Papavassiliu T et al. analyzing 69 patients with BrS found fatty infiltration in 15% of subjects [8]. A CMR study on 20 BrS patients observed high intramyocardial T1 signal similar to fat signal in 4% of individuals involved in the study [19]. On the other hand, Catalano O et al. performed CMR in 30 consecutive BrS patients, showing no fatty infiltration [16]. It has to be noted that the rate of tissue abnormalities reported in histological studies is significantly higher than that of CMR studies: this may reflect the difficulties of CMR in the detection of tissue pathological findings in the thin-walled RV myocardium. Therefore, it is likely that RV fibrosis, undetectable with current imaging technology, is also present in these patients and it could explain the BrS ECG phenotype [22]. However, data from the literature are not conclusive in regard to tissue characterization of BrS patients: fibrosis and fatty infiltration might be present only in the most severe cases. Further studies are needed in order to investigate tissue abnormalities in the spectrum of BrS.

### 3.6. Atrial Involvement

Considering the risk of atrial fibrillation in BrS patients, some studies assessed the presence of atrial abnormalities. A CMR comparison between BrS patients and healthy controls by Catalano et al. found no differences in atrial areas [16]. Bastiaenen et al. showed greater right atrial area in patients with BrS and SCN5A mutations [22]. Investigating various parameters, a study by Toh et al. found interesting results: the left atrium seemed to be abnormal in BrS patients. In particular, LA diameters and LA volume were significantly increased in patients with ventricular arrhythmias. LA volume was also increased in BrS patients with SCN5A mutation [35].

## 4. Conclusions and Future Directions

Imaging techniques allow an in-deep analysis of myocardial features. The present review of literature demonstrates the evidence of possible structural abnormalities in BrS: RV dilatation, abnormal RV ejection fraction, delayed contraction of RVOT, pathological strain values, and abnormal tissue characteristics. 

The impairment of these imaging parameters appears to be present, especially in BrS patients with genetic mutation. As a matter of fact, patients with BrS and a proven mutation in the sodium channel gene SCN5A showed significantly larger RV volumes and RVOT dimensions as well as lower RVEF and lower LV cardiac output than BrS patients without SCN5A mutation and healthy controls. A recent study highlighted that the abnormalities could change and evolve in the follow up [36]. Genetic background seems to be the main determinant for the extent of the electrophysiological abnormalities [37]. Our review of the literature in regard to imaging in BrS contributes to the understanding of the genetic determinants of BrS clinical expression. As a matter of fact, data from imaging studies could mean that a cardiomyopathy is behind clinical and electrocardiographic manifestations of BrS. These concepts provide a possible explanation for the varying degrees of disease expression. Future studies should focus on prospective imaging analysis of BrS patterns, genetic mutations, and prognostic implications. Furthermore, a potential relationship between imaging abnormalities (i.e., ventricular and atrial findings) and arrhythmias occurrence should be further investigated.

## Figures and Tables

**Figure 1 biology-12-00606-f001:**
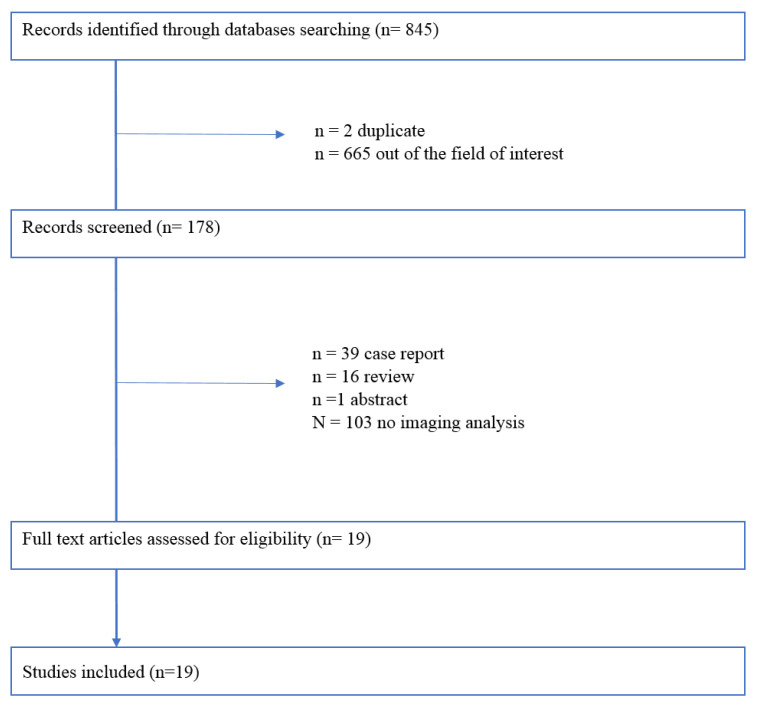
Outline of the search strategy.

**Figure 2 biology-12-00606-f002:**
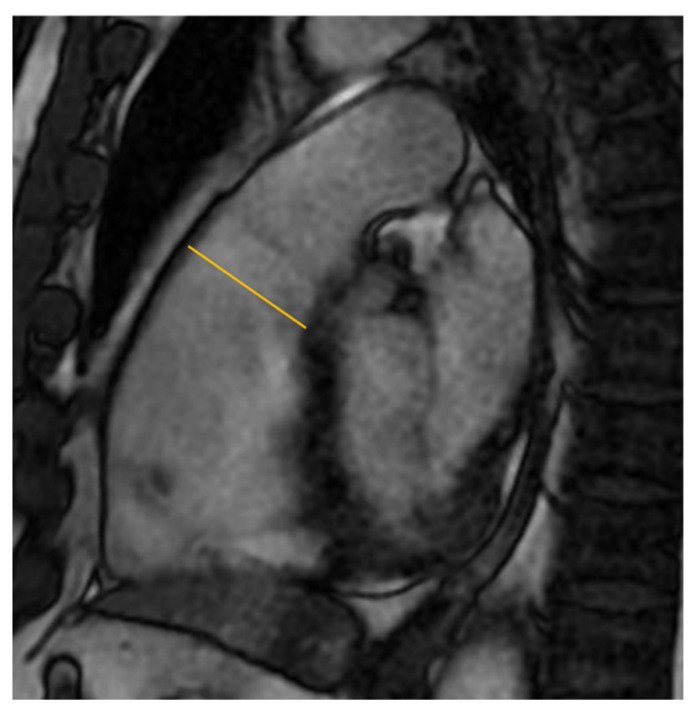
Right ventricular outflow tract: cine-CMR focused on dilated RVOT as measured in orange.

**Figure 3 biology-12-00606-f003:**
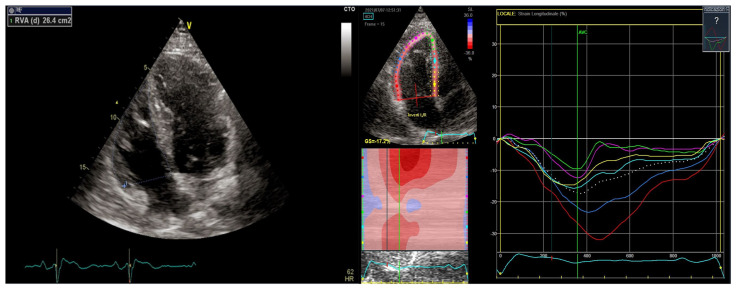
Right ventricular echocardiography showing a dilated RV and slightly abnormal RV strain values.

**Table 1 biology-12-00606-t001:** Main abnormalities detected by imaging techniques in Brugada Syndrome patients.

	Echocardiography	Cardiac MRI
**RV dilatation**	**+**	**+**
**RVEF abnormality**	**+**	**+**
**RV strain value**	**+**	**+**
**Delayed RV contraction**	**+**	**--**
**RV fatty infiltration**	**--**	**+**
**RV LGE**	**--**	**+**
**LVEF abnormality**	**+**	**+**
**Atrial enlargement**	**+**	**+**

Legend: LGE = late gadolinium enhancement; LVEF = left ventricular ejection fraction; RV = right ventricle; RVEF = right ventricular ejection fraction. + = parameter that can be assessed; - = parameter that can’t be assessed

**Table 2 biology-12-00606-t002:** Incidence of imaging abnormalities in BrS patients on the basis of the number of studies which detected the abnormality and data available about prognostic implications till now.

	Incidence	Prognostic Implication
**RV dilatation**	**+++**	
**RVEF abnormality**	**+++**	
**RV strain value**	**+++**	**+**
**Delayed RV contraction**	**+++**	**+**
**RV fatty infiltration**	**+**	
**RV LGE**	**+**	
**LVEF abnormality**	**+**	
**Atrial enlargement**	**++**	

Legend: LGE = late gadolinium enhancement; LVEF = left ventricular ejection fraction; RV = right ventricle; RVEF = right ventricular ejection fraction; + = low incidence (1 study); ++ = moderate incidence (2–4 studies); +++ = high incidence (at least 5 studies).

**Table 3 biology-12-00606-t003:** CMR findings about RV dimension in BrS.

Authors	Population	CMR Findings
Catalano O et al. [16]	30 BrS vs. 30 Hc	**↑** RV inflow tract diameter (*p* < 0.001)
		**↑** RV inflow tract area (*p* = 0.050)
		**↑** RVES volume (*p* = 0.031)
van Hoorn F et al. [17]	138 BrS vs. 18 Hc	**↑**RVOT diameter (*p* < 0.001)
		**↑** RVES volume (*p* < 0.001)
Rudic B et al. [18]	81 BrS vs. 30 Hc	**↑**RVOT diameter (*p* < 0.01)
		**↑** RVES volume (*p* < 0.01)
Papavassiliu T et al. [19]	20 BrS vs. 20 Hc	**↑** RV outflow area (*p* = 0.018)
		**↑** RVES volume (*p* = 0.3)
		**↑** RVED volume (*p* = 0.08)
Hohneck A et al. [20]	106 BrS vs. 25 Hc	**↑** RVES volume (*p* = 0.01)
		**↑** RVOT area (*p* < 0.0001)
Gray B et al. [21]	48 BrS vs. 48 Hc	**↑** RVOT volume (*p* < 0.0001)
Bastiaenen R et al. [22]	78 BrS vs. 78 Hc	**↑** RVES volume (*p* = 0.038)

Legend: BrS = Brugada Syndrome; Hc = Healthy controls; RV = right ventricular; RVOT = right ventricular outflow tract; RVES = right ventricular end systolic; RVED = right ventricular end diastolic.

**Table 4 biology-12-00606-t004:** CMR findings about RVEF and LVEF in BrS.

Authors	Population	RV CMR Findings	LV CMR Findings
Papavassiliu T et al. [19]	20 BrS vs. 20 Hc	trend ↓ RVEF (*p* = 0.06)	**x**
Catalano O et al. [16]	30 BrS vs. 30 Hc	RV motion abnormalities (*p* = 0.006)	**x**
		**↓** motion of RVOT (*p* = 0.032)	
Papavassiliu T et al. [8]	69 BrS (26 sBrS + 43 diBrS) vs. 30 Hc	↓ RVEF in sBrS (*p* < 0.05)	↓ LVEF in sBrS (*p* < 0.05)
Rudic B et al. [18]	81 BrS vs. 30 Hc	↓ RVEF (*p* < 0.01)	trend ↓ LVEF (*p* = 0.07)
Gray B et al. [21]	29 BrS vs. 29 Hc	↓ RVEF (*p* = 0.002)	x
		↓ motion of RVOT (*p* < 0.0001)	
Hohneck A et al. [20]	106 BrS (34 sBrS + 72 diBrS) vs. 25 Hc	↓ RVEF in sBrS vs. Hc (*p* < 0.0001)	**x**
van Hoorn F et al. [17]	138 BrS (40 SCN5A+ and 98 SCN5a−) vs. 18 Hc	**x**	↓ LVEF in SCN5A + (*p* < 0.001)
Tessa C. et al. [23]	29 BrS vs. 29 Hc	**x**	**x**

Legend: CMR = cardiac magnetic resonance; BrS = Brugada syndrome; Hc = healthy control; RVEF = right ventricular ejection fraction; RVOT = right ventricular outflow tact; LVEF = left ventricular ejection fraction.

## Data Availability

Not applicable.

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
