# Peer review of "Structural Abnormalities in Brugada Syndrome and Non-Invasive Cardiac Imaging: A Systematic Review"

_biology, 2023, doi:10.3390/biology12040606_

Round 1

Reviewer 1 Report

This is a well-written and well-referenced review. 

Minor comment

1. Expanding the "future direction" into a separate section might be interesting. It would add to the discussion, e.g., - the role of imaging in the ablation and risk stratification, - electroanatomical imaging.

2. It is also worth mentioning that the abnormalities may change and evolve during the follow-up. Please check this new interesting analysis: Isbister JC, Gray B, Offen S, Yeates L, Naoum C, Medi C, Raju H, Semsarian C, Puranik R, Sy RW. Longitudinal assessment of structural phenotype in Brugada syndrome using cardiac magnetic resonance imaging. Heart Rhythm O2. 2022 Oct 18;4(1):34-41. doi: 10.1016/j.hroo.2022.10.004. PMID: 36713046; PMCID: PMC9877394.

Author Response

MS Number: biology-2283579

Revision #1

Title: Structural abnormalities in Brugada syndrome and non-invasive cardiac imaging: a systematic review.

Corresponding Author: Dr Elisabetta Tonet

We would like to thank the editor and the reviewers for their time spent on our paper and their suggestions. We have modified our manuscript in order to respect these indications. We have aimed to incorporate most alterations.  Text changes in response to comments made by the reviewers are highlighted in blue.

Reviewer #1

Comment #1

Expanding the "future direction" into a separate section might be interesting. It would add to the discussion, e.g., - the role of imaging in the ablation and risk stratification, - electroanatomical imaging.

Reply to Comment #1

We thank the Reviewer #1 for this observation. This is the most important point for future research. Unfortunately, no evidence about imaging and electroanatomical mapping are available till now, but it could result of paramount importance in the future: for example, identifying LGE with CMR could be useful for ablation procedures. However, because of this gap of knowledge, we decided to maintain the current structure of the manuscript, no elements could be added as future directions so as to write a separate section.

Comment #2

It is also worth mentioning that the abnormalities may change and evolve during the follow-up. Please check this new interesting analysis: Isbister JC, Gray B, Offen S, Yeates L, Naoum C, Medi C, Raju H, Semsarian C, Puranik R, Sy RW. Longitudinal assessment of structural phenotype in Brugada syndrome using cardiac magnetic resonance imaging. Heart Rhythm O2. 2022 Oct 18;4(1):34-41. doi: 10.1016/j.hroo.2022.10.004. PMID: 36713046; PMCID: PMC9877394.

Reply to Comment #2

We thank the Reviewer #1 for this observation. In the new version of the manuscript we cited this paper.

See modified text: page 10, lines 326-327

A recent study highlighted that the abnormalities could change and evolve in the follow up [36].

See modified text: References section, page 12, lines 431-432

 Isbister JC, Gray B, Offen S et al, Longitudinal assessment of structural phenotype in Brugada syndrome using cardiac magnetic resonance imaging. Heart Rhythm O2. 2022 Oct 18;4(1):34-41.

Reviewer 2 Report

Accurate identification of imaging factors associated with symptoms, risk stratification, and prognosis in BrS patients which is the aim of the review could be clinically relevant.  However, the review does not clarify whether imaging can offer a robust contribution in this context. 

To start with I don’t agree that “Brugada syndrome has always been considered a purely electrical disease and imaging techniques do not currently play a specific role in the diagnosis of this arrhythmic syndrome.” In fact, BrS is diagnosed in patients without other heart disease. Phenocopies such as ARVC and pericarditis need to be excluded. This means that imaging, plays an important role at least in my clinic, for the clinical assessment of patients who are suspected of having BrS.  

In the abstract (row 27-28) is stated that “…features emerged more frequently in patients carrying the genetic mutation on the sodium voltage-gated channel α-subunit 5 (SCN5A) gene.” However, SCN5A is a complex gene harboring different variants with a considerable signal-to-noise ratio. While not one but several pathogenic SCN5A variants have been reported to cause BrS via loss-of-function mechanisms, other variants can cause LQT3, familial atrial fibrillation, Multifocal ectopic Purkinje-related premature contractions and, dilated cardiomyopathy (DCM) through different mechanisms. Therefore, studies presenting genetic data should be interpreted with great care. The present study lacks information regarding the certainty of genotypic findings in relation to BrS and to imaging phenotypes.  

The introduction should contain an estimate of the prevalence of BrS to enable a perception of the magnitude of the problem, i.e., a challenge such as risk stratification of patients with this channelopathy. BrS is a worldwide condition not confined only to Asian countries which is the impression given by the following “a prevalent geographic/ethnic distribution in Asian countries.” Also, BrS is not more common but more often reported to be manifested (i.e., expressed) in men. 

It was difficult to identify which articles were included in the study.  Also, I could not find reference 14 referred to in the text. The incidence of imaging abnormalities in BrS patients based on data from the included studies and given in Table 2 are graded as “+ = low incidence; ++ = moderate incidence; +++ = high incidence”.  But how was low, moderate, and high incidence defined? Statements such as “On one hand there are studies that support the absence of structural alterations in patients with BrS.” (row 126-127) should be backed by reference(s). 

I’m not sure I understand at all what the authors mean by “The first prospective analysis encompassing CMR in BrS patients was performed by Papavassiliu et al., who enrolled 20 BrS compared to matched controls [19].” (row 173-174). This is an example of what makes the paper difficult to read and to follow the thread. 

The authors should elaborate more on potential reasons for conflicting results. For example, “on one hand Von Hoorn et al. showed that patients carrying SCN5A mutation had significantly lower LVEF than non-SCN5A- BrS patients and healthy controls, without difference in RVEF [17].” Could it be so that the detected variants were in fact DCM-causing variants? The variant interpretation in the study by van Hoorn et al was made before 2015 when the first structured guidelines for the interpretation of sequence variants were published (PMID: 25741868). Many variants that were interpreted as pathogenic before 2015 have been reclassified as non-pathogenic. Please note that the name of the author is written incorrectly (row 207). What is a genetic pattern?  (row 213)

Differences are presented throughout the text but what are these clinically significant, what could be the clinical consequence? E.g. (row 273-276) “An analysis of differences among the three Brugada patterns showed that type 1 Brugada patients had a longer ejection time when compared with type 2 and 3 Brugada patients as well as controls.” How does this impact on symptoms, symptoms, risk stratification and prognosis?  Are there any type 2 and 3 Brugada patients?

Amending the text is necessary. Although seemingly minor things leave the reader with the overall impression of lack of thoroughness. 

1)         Row 13-15: the abbreviation BrS is used before it is explained: “Abstract: The aim of this review is to identify possible structural abnormalities of BrS and their 13 potential association with symptoms, risk stratification, and prognosis. (1) Background: Brugada syndrome (BrS) has always been considered a purely electrical disease …”.  

2)         Row 44 vs. row 50:  The references are not given in a consistent way “…Brugada syndrome. [2, 3].”   vs. “…sodium-channel blockers [1, 4-5].” 

3)         Row 116, 121, 165, 219: ”Legenda” ???

4)         Row 161 vs. 245 vs. “Figure 2. Right ventricular…”” Figure 3. Right …” 

5)         Row 247 vs. 267 “Mitroi et al. ..” vs. ” Mitroi C et al found”

6)         Row 319: the refererad Figure 4 is absent in the manuscript.

In summary, this review does not clarify if and how imaging contributes in a robust way to prediction of symptoms, risk stratification and prognosis in BrS patients. 

Author Response

Reviewer #2

Comment #1

Accurate identification of imaging factors associated with symptoms, risk stratification, and prognosis in BrS patients which is the aim of the review could be clinically relevant.  However, the review does not clarify whether imaging can offer a robust contribution in this context. 

Reply to comment #1

We thank the Reviewer #2 for his/her comment. We apologize if the expectations of the Reviewer #2 were not satisfied. Our review aimed to report all evidence available till now about imaging in BrS: on the basis of our analysis of data from literature we conclude that BrS could be characterized by structural abnormalities (highlighted in our manuscript), but further studies and prospective assessment are needed in order to confirm the robust contribution of imaging technique that is currently only hypothesized.

Comment #2

To start with I don’t agree that “Brugada syndrome has always been considered a purely electrical disease and imaging techniques do not currently play a specific role in the diagnosis of this arrhythmic syndrome.” In fact, BrS is diagnosed in patients without other heart disease. Phenocopies such as ARVC and pericarditis need to be excluded. This means that imaging, plays an important role at least in my clinic, for the clinical assessment of patients who are suspected of having BrS.  

Reply to Comment #2

We thank again the Reviewer #2 for this observation. The Reviewer #2 is right. As a matter of fact, imaging techniques are currently used for exclusion of other possible diagnosis. We highlight this concept in the arranged version of the manuscript.

See modified text: page 2, lines 60-63

Initially considered as a purely electrical disease, using cardiac imaging just to exclude other cardiomyopathies, some authors have recently hypothesized the presence of structural and functional abnormalities, mostly located in the right ventricle (RV) and, specifically, in the outflow tract

Comment #3

In the abstract (row 27-28) is stated that “…features emerged more frequently in patients carrying the genetic mutation on the sodium voltage-gated channel α-subunit 5 (SCN5A) gene.” However, SCN5A is a complex gene harboring different variants with a considerable signal-to-noise ratio. While not one but several pathogenic SCN5A variants have been reported to cause BrS via loss-of-function mechanisms, other variants can cause LQT3, familial atrial fibrillation, Multifocal ectopic Purkinje-related premature contractions and, dilated cardiomyopathy (DCM) through different mechanisms. Therefore, studies presenting genetic data should be interpreted with great care. The present study lacks information regarding the certainty of genotypic findings in relation to BrS and to imaging phenotypes.  

Reply to Comment #3

The Reviewer #2 is again right. However, the present review aimed to report structural abnormalities highlighted in literature in BrS patients, without discussing the truthfulness and variants of genetic mutations: genetic data reported by the analyzed studies were considered as they were reported and published by impacted journals.

Comment #4

The introduction should contain an estimate of the prevalence of BrS to enable a perception of the magnitude of the problem, i.e., a challenge such as risk stratification of patients with this channelopathy. BrS is a worldwide condition not confined only to Asian countries which is the impression given by the following “a prevalent geographic/ethnic distribution in Asian countries.” Also, BrS is not more common but more often reported to be manifested (i.e., expressed) in men. 

Reply to Comment #4

We thank the Reviewer #2 for this interesting observation. We arranged the manuscript accordingly.

See modified text: page 1, lines 39-41

The worldwide prevalence of BrS is estimated to be 0.05%, it is more common in young male with a prevalent geographic/ethnic distribution in Asian countries [1, 2].

Comment #5

It was difficult to identify which articles were included in the study.  Also, I could not find reference 14 referred to in the text. The incidence of imaging abnormalities in BrS patients based on data from the included studies and given in Table 2 are graded as “+ = low incidence; ++ = moderate incidence; +++ = high incidence”.  But how was low, moderate, and high incidence defined? Statements such as “On one hand there are studies that support the absence of structural alterations in patients with BrS.” (row 126-127) should be backed by reference(s). 

Reply to Comment #5

We thank the Reviewer #2 for this comment. We apologize for not being clear enough about articles included in the anlaysis. We tried to better explain it in the new version of the manuscript.

See modified text: page 4, lines 123-125

Legenda: LGE=late gadolinium enhancement; LVEF=left ventricular ejection fraction; RV=right ventricle; RVEF=right ventricular ejection fraction; + = low incidence (1 study); ++ = moderate incidence (2-4 studies); +++ = high incidence (at least 5 studies)

See modified text: page 4, lines 128-129

On one hand there are studies that support the absence of structural alterations in patients with BrS [12].

Comment #6

I’m not sure I understand at all what the authors mean by “The first prospective analysis encompassing CMR in BrS patients was performed by Papavassiliu et al., who enrolled 20 BrS compared to matched controls [19].” (row 173-174). This is an example of what makes the paper difficult to read and to follow the thread. 

Reply to Comment #6

We apologize for not being clear enough. W meant that the first anakyss of BrS individuals with CMR was performed by Papavassiliu.

Comment #7

The authors should elaborate more on potential reasons for conflicting results. For example, “on one hand Von Hoorn et al. showed that patients carrying SCN5A mutation had significantly lower LVEF than non-SCN5A- BrS patients and healthy controls, without difference in RVEF [17].” Could it be so that the detected variants were in fact DCM-causing variants? The variant interpretation in the study by van Hoorn et al was made before 2015 when the first structured guidelines for the interpretation of sequence variants were published (PMID: 25741868). Many variants that were interpreted as pathogenic before 2015 have been reclassified as non-pathogenic. Please note that the name of the author is written incorrectly (row 207). What is a genetic pattern?  (row 213)

Reply to Comment #7

Thank you for this comment. The observations are right. However, the aim of our manuscript was not to dispute data published by other authors.

We arranged the rows observed in this comment.

See modified text: page 7, line 216

genetic typing

Comment #8

Differences are presented throughout the text but what are these clinically significant, what could be the clinical consequence? E.g. (row 273-276) “An analysis of differences among the three Brugada patterns showed that type 1 Brugada patients had a longer ejection time when compared with type 2 and 3 Brugada patients as well as controls.” How does this impact on symptoms, symptoms, risk stratification and prognosis?  Are there any type 2 and 3 Brugada patients?

Reply to Comment #8

We thank the Reviewer #2 for this observation. Unfortunately, clinical implications and prognostic meaning of imaging abnormalities could not be more in-deep reported in this manuscript because there is no evidence in literature till now. The few data about prognostic implications have been reported.

Comment #9

Amending the text is necessary. Although seemingly minor things leave the reader with the overall impression of lack of thoroughness. 

1)         Row 13-15: the abbreviation BrS is used before it is explained: “Abstract: The aim of this review is to identify possible structural abnormalities of BrS and their 13 potential association with symptoms, risk stratification, and prognosis. (1) Background: Brugada syndrome (BrS) has always been considered a purely electrical disease …”.  

2)         Row 44 vs. row 50:  The references are not given in a consistent way “…Brugada syndrome. [2, 3].”   vs. “…sodium-channel blockers [1, 4-5].” 

3)         Row 116, 121, 165, 219: ”Legenda” ???

4)         Row 161 vs. 245 vs. “Figure 2. Right ventricular…”” Figure 3. Right …” 

5)         Row 247 vs. 267 “Mitroi et al. ..” vs. ” Mitroi C et al found”

6)         Row 319: the refererad Figure 4 is absent in the manuscript.

In summary, this review does not clarify if and how imaging contributes in a robust way to prediction of symptoms, risk stratification and prognosis in BrS patients. 

Reply to Comment #9

We apologize for not being clear enough. We arranged the manuscript according to the observations.

Reviewer 3 Report

Thank you for the opportunity to review this manuscript entitled “Structural abnormalities in Brugada Syndrome and non-invasive cardiac imaging: a systemic review”. This is a review summarizing published literature on imaging findings and structural abnormalities in patients with Brugada syndrome. The manuscript  is clinical relevant and reviews an interesting and relatively novel concept in patients with this condition, which was previously thought to occur in the absence of structural abnormalities. Overall the review is a good addition to available literature on the topic. 

I have the following specific comments:

1. In Table 2, the incidence of imaging findings is quantified as +,++, or +++. Is it possible to a little more quantification to this, le, what incidence range of these findings does each category represent? 

2. Some abbreviations used were confusing in the text (eg sBrS and diBrS, as well as CMR-FT). These abbreviations should be limited when possible. 

3. In section 3.4 on Contraction delay there was brief discussion on potential implications of the imaging findings on pt outcomes. This is potentially very useful information- perhaps a section could be added commenting on potential implications of the various imaging findings on prognosis, if any such implications could be generated from available literature. 

Author Response

Reviewer #3

Thank you for the opportunity to review this manuscript entitled “Structural abnormalities in Brugada Syndrome and non-invasive cardiac imaging: a systemic review”. This is a review summarizing published literature on imaging findings and structural abnormalities in patients with Brugada syndrome. The manuscript  is clinical relevant and reviews an interesting and relatively novel concept in patients with this condition, which was previously thought to occur in the absence of structural abnormalities. Overall the review is a good addition to available literature on the topic. 

We thank the Reviewer #2 for his/her considerations.

Comment #1

In Table 2, the incidence of imaging findings is quantified as +,++, or +++. Is it possible to a little more quantification to this, le, what incidence range of these findings does each category represent? 

Reply to Comment #1

We thank the Reviewer #2 for this suggestion. We better defined the incidence.

See modified text: page 4, lines 123-125

Legenda: LGE=late gadolinium enhancement; LVEF=left ventricular ejection fraction; RV=right ventricle; RVEF=right ventricular ejection fraction; + = low incidence (1 study); ++ = moderate incidence (2-4 studies); +++ = high incidence (at least 5 studies)

Comment #2

Some abbreviations used were confusing in the text (eg sBrS and diBrS, as well as CMR-FT). These abbreviations should be limited when possible. 

Reply to Comment #2

We thank the Reviewer #2 for this observation. We apologize for not being clear enough. We arranged the new version of the manuscript according to this suggestion.

See modified text: page 7, lines 202-207

In a recent study by Hohnek et al, CMR and CMR feature tracking analysis were performed on a cohort of BrS patients, both spontaneous BrS and drug induced BrS patterns, compared with matched healthy controls. Authors underlined a trend to lower biventricular EF in patients with spontaneous BrS pattern when compared with subjects with drug induced BrS pattern and healthy controls, respectively, even if the difference did not reach statistical significance

See modified text: page 8, lines 254-261

A prospective study by Hohnek et al, evaluated the prognostic impact of CMR feature tracking in a well characterized cohort of BrS patients, to detect subclinical impairment and identify those with higher risk for arrhythmogenic events. CMR feature tracking showed remarkable differences in RV strain parameters between healthy controls and patients with spontaneous BrS pattern, while conventional CMR measures were comparable. On the contrary, no CMR feature tracking difference was found among patients with drug induced BrS pattern and healthy population. Considering follow-up, major adverse event occurred mainly in patients with spontaneous BrS pattern.

Comment #3

In section 3.4 on Contraction delay there was brief discussion on potential implications of the imaging findings on pt outcomes. This is potentially very useful information- perhaps a section could be added commenting on potential implications of the various imaging findings on prognosis, if any such implications could be generated from available literature. 

Reply to Comment #3

We thank the Reviewer #2 for this comment. This is an important point. Unfortunately very few data are now available about prognostic role of imaging parameters. These data have been incorporated in the manuscript in each section and in the new version of the manuscript we added a column to the Table 2 about prognostic implication data available till now.

See modified text: page 4, Table 2

Table 2. Incidence of imaging abnormalities in BrS patients on the basis of the number of studies which detected the abnormality and data available about prognostic implications till now.

Round 2

Reviewer 2 Report

The authors addressed the comments adequately; hence the manuscript has improved. I have no further comments.